# The Hydro-Isostatic Rebound Related to Megalake Chad (Holocene, Africa): First Numerical Modelling and Significance for Paleo-Shorelines Elevation

**Anthony Mémin [1], Jean-François Ghienne [2], Jacques Hinderer [2], Claude Roquin [2] and Mathieu Schuster [2,*]**

[1]  Université Côte d'Azur, CNRS, Observatoire de la Côte d'Azur, IRD, Géoazur, 06560 Valbonne, France; memin@geoazur.unice.fr

[2]  Université de Strasbourg, CNRS, Institut de Physique du Globe de Strasbourg UMR 7516, 67084 Strasbourg, France; ghienne@unistra.fr (J.-F.G.); jhinderer@unistra.fr (J.H.); claude.roquin@sfr.fr (C.R.)

*  Correspondence: mschuster@unistra.fr

**Abstract:** Lake Chad, the largest freshwater lake of north-central Africa and one of the largest lakes of Africa, is the relict of a giant Quaternary lake (i.e., Megalake Chad) that developed during the early- to mid-Holocene African Humid Period. Over the drylands of the Sahara Desert and the semi-arid Sahel region, remote sensing (optical satellite imagery and digital elevation models) proved a successful approach to identify the paleo-shorelines of this giant paleo-lake. Here we present the first attempt to estimate the isostatic response of the lithosphere due to Megalake Chad and its impact on the elevation of these paleo-shorelines. For this purpose, we use the open source TABOO software (University of Urbino, Italy) and test four different Earth models, considering different parameters for the lithosphere and the upper mantle, and the spatial distribution of the water mass. We make the simplification of an instantaneous drying-up of Megalake Chad, and compute the readjustment related to this instant unload. Results (i.e., duration, amplitude, and location of the deformation) are then discussed in the light of four key areas of the basin displaying prominent paleo-shoreline morpho-sedimentary features. Whatever the Earth model and simplification involved in the simulations, this work provides a strong first-order evaluation of the impact on hydro-isostasy of Megalake Chad. It demonstrates that a water body similar to this megalake would induce a significant deformation of the lithosphere in the form of a vertical differential uplift at basin-scale reaching up to 16 m in the deepest part of the paleo-lake, and its shorelines would then be deflected from 2 m (southern shorelines) to 12 m (northern shorelines), with a maximum rate of more than 1 cm y$^{-1}$. As such, any future study related to the paleo-shorelines of Megalake Chad, should integrate such temporal and spatial variation of their elevations.

**Keywords:** green Sahara episode; African humid period; lithosphere; uplift; readjustment; isostasy; continental hydrosystems; remote sensing; beach ridges; TABOO post-glacial rebound calculator

## 1. Introduction

Amongst the catalogue of potential links between climate change and deep Earth processes, the isostatic response of the lithosphere to loading or unloading by ice sheets or water bodies is surely the one with the most significant impact on continental surface changes and on related sedimentary systems reorganization. Everywhere the Last Glacial Maximum ice sheets have severely deflected the continental lithosphere, the patterns of the post-glacial isostatic rebound and of its geomorphic signatures (e.g., raised beaches, deltas, and overflow location) [1–5] show uplift rates in the 1–10 cm yr$^{-1}$ range, resulting in total vertical displacement commonly in excess of 500 m. Postglacial rebound has been

routinely used to extract deep Earth properties such as mantle viscosity [6–8]. Sophisticated models of former ice-sheets extent and volume also have to account for the isostatic response of the lithosphere [9]. Though involving much smaller deflections, hydro-isostasy is also significant [10]. For instance, accounting for it is crucial when estimating ice sheet volumes from observed relative sea-level fluctuations [11]. Similarly, the understanding of hydro-isostatic processes in lake studies is also of prime importance, as far as large inland water bodies experiencing significant water-level fluctuations are concerned (e.g., lakes Agassiz, Bonneville, and Lahontan). Indeed, vertical deflection in meters to tens of meters has been demonstrated to deform ancient lake shorelines [12–27].

In the present study, we consider Megalake Chad (MLC), the largest lake that developed in the Sahara–Sahel region during the Holocene [28,29]. Following pioneering studies [30–34], series of past shoreline elevations based on present-day altitudes given by existing digital elevation models such as SRTM [35] are available from the literature [28,29,36–43], notwithstanding (i) the difficulty to precisely point a water-level on the basis of complex littoral landforms [29,42], and (ii) potential deformation of the former shorelines since the drying-up of the megalake. However, considering the remarkable dimensions of this immense paleo-lake that expanded over almost 8° of latitude and 5° of longitude (water-surface: ~350,000 km$^2$; maximal water-depth: ~160 m), and by comparison with other case studies [24], this water body most likely forced a significant deformation of the lithosphere. In order to quantitatively estimate the importance of this deformation, we model here for the first time the vertical deflection and pattern of the hydro-isostatic readjustment caused by the flooding (i.e., loading) and then the drying-up (i.e., unloading) of this large paleo-lake, provided that the lithosphere deflection reached the isostatic equilibrium before it started to dry up.

Our ambition is not to characterize deep Earth rheology in this intracratonic part of Africa [44–46]—a standard mantle viscosity has been introduced in the model—but instead to discuss to which extent accounting for hydro-isostasy is crucial when deciphering former shoreline elevations and interpreting the late Quaternary MLC development. The model suggests that up to 16 m of vertical displacement may have occurred during the upper Holocene in the Chad Basin, with a potential deflection of the MLC shorelines in the 2–12 m range from south to north. Accounting for such vertical variations is crucial for the understanding of elevation distributions tied to a variety of morpho-sedimentary features.

## 2. Materials and Methods

### 2.1. Case Study: Megalake Chad

Megalake Chad (Figure 1) is a huge extinct lake of more than 350,000 km$^2$ that developed following the Holocene climatic optimum called the African Humid Period, also known for that part of Africa as the Green Sahara Episode [42]. This climate event was forced by orbital parameters, which led to a northward shift of the Intertropical Convergence Zone (ITCZ) and a correlative increase of monsoonal rainfall over the Sahel and Sahara ([47,48], and references herein). As a consequence, extinct river systems were temporarily reactivated [49,50], leading to the development of wetlands and of numerous lakes in the Sahara–Sahel, among which is Megalake Chad [51]. These major paleoenvironmental changes have in turn favored the human occupation of the Sahara and the development of diverse social-ecological systems [52–56]. A set of prominent littoral morpho-sedimentary structures, best observed from remote sensing, are organized all around the modern Lake Chad basin. They are distributed roughly at constant elevation (ca. 325 ± 10 m above sea level) and form a few thousands of kilometers long loop. All together these paleo-shorelines allow to firmly map the outline of this former giant single water body [29]. Those massive and diversified clastic coastal landforms mainly result from wind-driven hydrodynamics [29,41,42,57]. They include remarkable series of beach ridges and swales, spits, wave-ravinement surfaces, and lobate to cuspate deltas. Several portions of these paleo-shorelines have been already described and commented in former papers [29,41,42], and serve hereafter as key areas to assess the results of the modelling performed in this study.

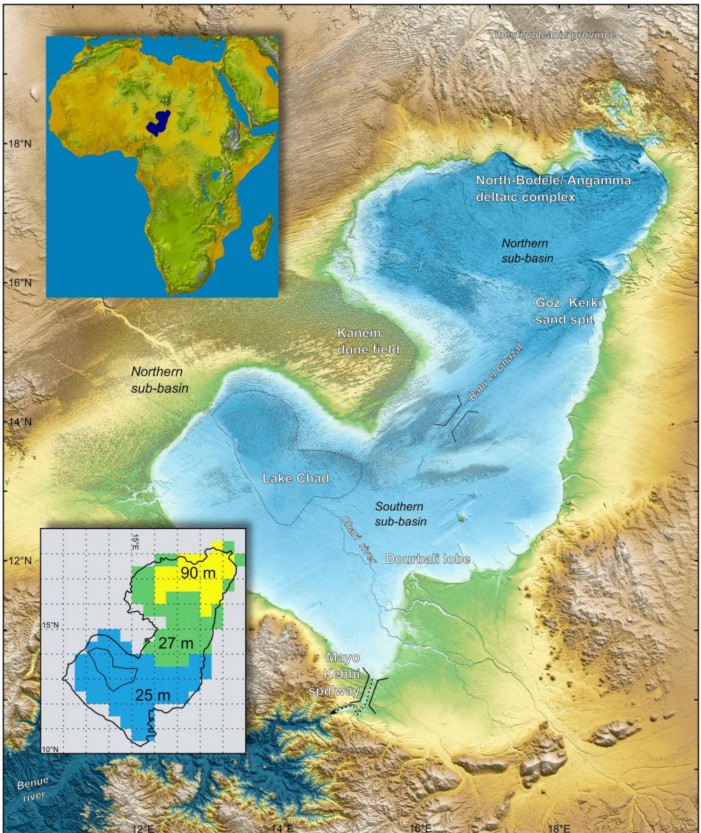

**Figure 1.** Digital elevation model (SRTM3) of the Chad Basin showing the paleo-extent of Megalake Chad (in light blue), the outline of the Lake Chad in the 1960s (dotted line), and the location of the main areas mentioned in the main text (for more details see [29,40–42]). Upper-left inset: extent of Megalake Chad compared to Africa. Lower-left inset: approximation used to compute the hydro-isostatic rebound related to the drying up of the MLC (the black line represent the 325 m-level line; note the three distinct sub-divisions used to model the load, values refers to their respective mean depths).

The basin of Lake Chad is endoreic up to the elevation of the topographic sill of the Mayo Kebbi (325 m asl) that connects to the south with the adjacent drainage basin of the Benue and Niger rivers (Figure 1). As a consequence, the theoretical elevation that a lake highstand can reach in this basin is given by the elevation of this sill. The real elevation likely being some meters above, considering an unknown water flow thickness across the outflow during MLC events, as well as an unknown depth of fluvial incision there over time. Last, it is worth noting for this study that Megalake Chad is divided into two domains, one to the south and another one to the north [37]. The southern sub-basin is the shallowest with a maximum depth of about 40 m, and the northern sub-basin is the deepest with a maximum depth of about 160 m. This peculiar configuration implies a non-uniform load by the water mass resulting from the development of MLC.

From a geomorphological point of view, and irrespective of their absolute elevations, the MLC shoreline features are typically organized in four sets of structures. From the higher and older to the lower and younger features are (i) partly degraded features, sometimes overlain by late Pleistocene (Kanemian) aeolian sand dunes; (ii) series of shoreline features, essentially at the same elevation within a particular place, marking the relatively long-term progradation of a strandplain either in delta settings or associated with sand spits; (iii) one or two singular beach ridges post-dating and lying 2–3 m below the last strandplain structures, with which they appear unconformable in map view; they are thus considered as originating from a later and relatively short-lived lacustrine highstand; and (iv) series of isolated and regressive beach ridges at largely lower elevations marking later successive positions of the shorelines marking the inexorable climate-driven shrinking of the lake.

Series of absolute ages ([29,43,58] and references therein) allow to constrain the MLC events in the 5–12 ky BP time range. The more degraded features are considered to be the older ones, while the isolated regressive beach ridges are the younger ones, with a lake still-stand recognized around 3 ky BP. Most of the 5–12 ky BP timespan is regarded as corresponding to the accretion of the strandplains, yet distinct associated elevation ranges can be determined (330–333 m, 333–337 m, and 347–340 m in the Dourbali lobe of the Chari river delta, the Goz Kerki sand spit, and the north-Bodélé/Angamma deltaic complex, respectively). Discrete phases of lower lake-level elevations resulted in internal unconformities, the latter of which separating the strandplain and the couple of later singular beach ridges preceding the final drying-up of the lake that occurred after 5 ky as a consequence of the termination of the African Humid Period [29].

## 2.2. Modelling

We use the open source TABOO software [59,60] to compute the vertical displacements, $U$, and the change in the geoid height, $N$, induced by the visco-elastic deformation due to the past changes in the water-level of MLC. The modelling procedure is based on the Green functions formalism introduced by Farrell [61] and extended by Peltier [62]. Vertical displacement and geoid change Green functions represent the response of the Earth to a unit load applied on its surface. After discretization of the MLC (i.e., the load), we computed the Earth surface deformation induced by the entire loading history (i.e., changes in the water-level of the MLC) by convolving Green functions with the space and time variations of every load elements. Green functions are computed assuming four incompressible, self-gravitating, non-rotating, visco-elastic, and spherically symmetric Earth models (M1–M4) with three layers between the core-mantle boundary and the surface. These layers correspond to the lithosphere, the upper and lower mantle (Table 1). The radii of the core-mantle and of the upper-lower mantle boundaries are 3480 and 5701 km, respectively. The radius of the boundary between the upper mantle and the lithosphere ranges between 6251 and 6321 km, depending on the lithosphere thickness. The elastic properties of the Earth models are volume averages of the preliminary reference Earth model—PREM [63]. We use the Maxwell rheology to model the visco-elastic deformation.

**Table 1.** Characteristics of the four Earth models (M1 to M4) used here to compute vertical displacements and changes in the geoid height induced by the drying-up of the MLC. Notation: L corresponds to lithosphere thickness, T and η correspond to mantle thickness and viscosity, respectively. The core-mantle and upper-lower mantle boundaries are at depths of 2891 km and 670 km, respectively.

| Earth Models | Lithosphere | Upper Mantle | | Lower Mantle | |
|:---:|:---:|:---:|:---:|:---:|:---:|
| | L (km) | T (km) | η ($10^{20}$ Pa·s) | T (km) | η ($10^{21}$ Pa·s) |
| **M1** | 100 | 570 | 5 | 2221 | 2.7 |
| **M2** | 100 | 570 | 0.5 | 2221 | 2.7 |
| **M3** | 100 | 570 | 0.05 | 2221 | 2.7 |
| **M4** | 50 | 620 | 0.05 | 2221 | 2.7 |

In order to characterize the hydro-isostatic rebound of the MLC we use the 325 m contour line as an approximate, but realistic, paleo-lake water-level. Based on the peculiar bipartite bathymetric configuration of the lake basin, the modelled lake is here divided into a northern and a southern sub-basin of ca. 183,000 and 165,000 km$^2$, respectively. The former being further subdivided into a shallow domain (above the 275 m contour line; paleo-depth <50 m) and a deep domain (below the 275 m contour line; paleo-depth >50 m). This subdivision basically accounts for the strong asymmetry of the shape of the northern sub-basin that shows greater depths immediately off the northern lacustrine shore (Figure 1). With estimated volumes of 9850 and 4550 km$^3$ in the northern and southern sub-basins, we deduce initial mean water heights of 90 m, 27 m, and 25 m for the deep and shallow domains of the northern sub-basin, and the southern sub-basin, respectively. The vertical displacement and the change in the geoid height were then modelled by filling each sub-basin and domain with water-load elements

of $0.5° \times 0.5°$ (Figure 1), assuming that the lithosphere deflection reached the isostatic equilibrium before the drying-up of the lake started, and that the MLC dried up instantaneously. These two simplifications imply that uplifts, which are calculated here up to 5 ky after the lake disappearance, have to be regarded as maximum values for the hydro-isostatic rebound, both within and around the area that was formerly flooded by MLC.

Different values of lithosphere thickness and upper mantle viscosity (Table 1) are used here to assess the effects of the radial structure of the Earth rheology on the vertical displacement and the change in the geoid height [8]. Model M1 has a lithosphere thickness of 100 km, a lower mantle viscosity of $2.7 \times 10^{21}$ Pa·s, and an upper mantle viscosity of $5 \times 10^{20}$ Pa·s. Such mantle viscosities correspond to a volume-averaged estimate of the viscosity structure (VM2) used by Peltier [64] for glacial-isostatic adjustment studies. Models M2 and M3 show the effects of considering lower estimates of upper mantle viscosities ($5 \times 10^{19}$ and $5 \times 10^{18}$ Pa·s). In an attempt to account for recent volcanic activity in the Tibesti massif area [65], which is associated to a notably thinned lithosphere to the north of the Chad Basin [46], a fourth configuration, model M4, has been also modeled, which only differs from model M3 by a twice as thin lithosphere (i.e., 50 km).

## 3. Results

Results of the numerical simulations for the four models are here displayed on four maps (Figure 2), representing the cumulated difference of the vertical displacement and the change in the geoid height after 5 ky of load release and isostatic readjustment. Change in the geoid height (some tens of centimeters) reveals to be very minor compared to the vertical displacement (several meters), and as such it does not contribute significantly to uplift amplitudes. In addition, the uplift curves through time at a specific location (north-Bodélé/Angamma deltaic complex area; Figure 3), shows that the four models do not reach the equilibrium at the same time.

For model M1, the isostatic adjustment is not fully achieved after 5 ky, and rebound is then only partial. Model M1 shows maximum uplift values of 7 m in the center of the northern sub-basin, in the 4–6 m range for the paleo-shorelines of the northern sub-basin, and of less than 3 m along most of the paleo-shorelines of the southern sub-basin.

Models M2 and M3 show very minor differences in map view (Figure 2) since in both cases the isostatic adjustment is virtually fully achieved after 5 ky (Figure 3), owing to reduced upper mantle viscosities. Both models thus show slightly greater uplift amplitudes, reaching up to 10 m in the northern sub-basin, and in the 2–5 m range along the paleo-shorelines of the southern sub-basin. The rate of uplift in the initial thousands of years of the readjustment is the only significant difference between models M2 and M3.

Model M4, in which the adjustment was almost terminated in less than 1 ky, shows uplift amplitudes at 5 ky that are two times greater than those of model M1, up to 16 m in the center of the northern sub-basin. The effect of a thinned lithosphere is especially noticeable in the northern sub-basin, where ca. 12 m of hydro-isostatic uplift is modeled for the northernmost paleo-shorelines of the north-Bodélé/Angamma deltaic complex area, and uplift of more than 8 m for the paleo-shorelines to the north and east of the northern sub-basin. Furthermore, along-strike variation in uplift amplitudes is significant, with up to 5 m of elevation difference within individual 100 km long segments of shoreline. Finally, it is worth noticing (Figure 3) that only with an Earth model approaching model M1 and with an instantaneous unload, the hydro-isostatic rebound in the Chad Basin would still be effective today.

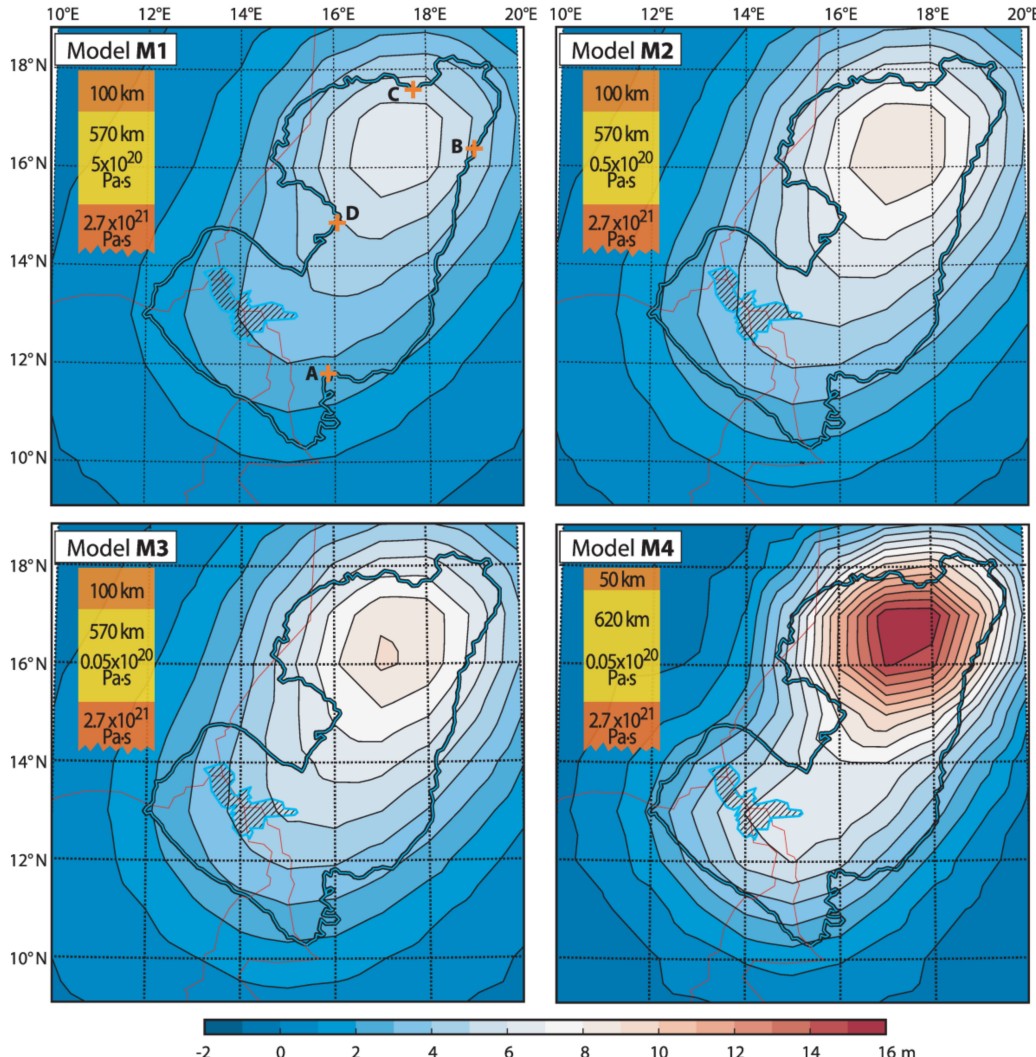

**Figure 2.** Maps representing the amplitude of the hydro-isostatic rebound induced by the drying-up of Megalake Chad, according to the four Earth models (M1–M4; Table 1). Contour lines are equally spaced at a 1 m interval.

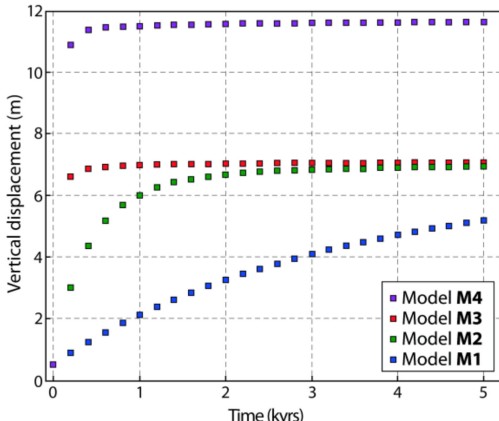

**Figure 3.** Time variation of the vertical displacement subsequent to the instantaneous disappearance of Megalake Chad as modeled in the area of the North-Bodélé/Angamma deltaic complex and according to the four Earth models (M1–M4). In this coastal area showing the greatest hydro-isostatic rebound amongst the four studied areas, readjustment is quite rapidly fully achieved considering models M2, M3, and M4, but would be currently still active if model M1 was accounted for.

## 4. Discussion

### 4.1. Mayo Kebbi Spillway

The maximum water-level reached by MLC was controlled by an outflow channel located in the Mayo Kebbi region, south of modern the Lake Chad (Figure 1), which the current elevation is 325 m asl according to the SRTM1 digital elevation model. Whatever the chosen model of hydro-isostatic rebound the corrected elevation of this threshold is ca. 323–324 m, considering in average a 1.5 ± 0.5 m isostatic uplift for the related area (Figure 2).

Three areas exhibiting prominent shore-related morpho-sedimentary structures that clearly mark the local position of the paleo-shorelines are reviewed hereafter in order to examine to which degree, accounting for the hydro-isostatic rebound helps to understand the distribution of MLC shoreline elevations, these latter being interestingly not uniform from south to north of the basin. Table 2 provides elevation ranges for the strandplain and for the late, singular beach ridges in the areas of the Dourbali lobe of the Chari river delta, of the north-Bodélé/Angamma deltaic complex, and of the Goz Kerki sand spit. In each case, with no isostatic correction, shoreline features are clearly positioned at too higher and different elevations relative to the Mayo Kebbi spillway. It is worth noting that in each case the fit becomes better as soon as an isostatic correction is considered, and discrepancies from place to place are then reduced. However, in both the Dourbali lobe of the Chari river delta and the north-Bodélé/Angamma deltaic complex case studies, even maximal corrections (e.g., the ones inferred from model M4), do not allow to reconcile observed present-day elevations with corrected elevations if the lake-level was exactly adjusted to the elevation of the Mayo Kebbi spillway (Table 2).

**Table 2.** Elevations of the paleo-shorelines of Megalake Chad at three key-areas, elevation correction according to the different Earth models relative to the Mayo Kebbi spillway, and expected present-day elevation of past MLC shorelines for a MLC set at the elevation of this spillway.

| | | Dourbali Lobe of the Chari River Delta | North-Bodélé/Angamma Deltaic Complex | Goz Kerki Sand Spit |
|---|---|---|---|---|
| **Elevation of the strandplain** | | 330 to 333 m asl | 340 to 347 m asl | 332 to 337 m asl |
| **Elevation of the late, singular beach ridges** | | poorly defined, no value | ca. 338 m | ca. 330 m |
| **Differential isostatic uplift relative to Mayo Kebbi (±1 m)** | M1 | 1.5 m | 4.5 m | 4 m |
| | M3 | 2 m | 6 m | 4.5 m |
| | M4 | 3.5 m | 10.5 m | 8.5 m |
| **Expected present-day elevation of past MLC shorelines if once adjusted to that of the spillway (325 m asl)** | M1 | 326.5 m | 329.5 m | 329 m |
| | M4 | 328.5 m | 335.5 m | 333.5 m |
| **Residual mismatch** | M1 | >3.5 m | >10.5 m | >3 m |
| | M4 | >1.5 m | >4.5 m | potentially no mismatch |

Rather than testing other, more or less unrealistic, Earth models that would produce greater isostatic uplifts, we hypothesize that this residual mismatch mainly originates from a water-level that was not exactly fitted to the spillway crest elevation, but that was slightly higher, owing to the inherent thickness of the massive water overflow that was once exiting the lake through the Mayo Kebbi spillway during MLC events. For each of the three study areas, the differential uplift relative to this spillway (Table 2) allows the present-day elevation of the former lake shoreline to be predicted for distinct Earth models and for three hypothesized thickness for the water stream, which were arbitrarily taken at 0 m, 4 m, and 8 m above the present-day elevation of the spillway (Table 3).

**Table 3.** Elevations of the paleo-shorelines of Megalake Chad at three key-areas and expected present-day shoreline elevation according to the Earth models and accounting for three different overflow thickness above the Mayo Kebbi spillway. The ca. 0.5–1 m correction related to the resulting additional water load is here neglected (see Figure 4). Grey cells correspond to reconstructed elevations within the range of measured paleo-shoreline elevation.

| | Overflow Thickness Above the Spillway | Dourbali Lobe of the Chari River Delta | North-Bodélé/Angamma Deltaic Complex | Goz Kerki Sand Spit |
|---|---|---|---|---|
| **Elevation of the strandplain** | - | 330–333 m asl | 340–347 m asl | 332–337 m asl |
| **Elevation of the late, singular beach ridges** | - | poorly defined, no value | ca. 338 m | ca. 330 m |
| **Model M1** | 0 m | 326.5 m | 329.5 m | 329 m |
| Expected present-day elevation of past | 4 m | 330.5 m | 333.5 m | 333 m |
| MLC shorelines | 8 m | 334.5 m | 337.5 m | 337 m |
| **Model M3** | 0 m | 327 m | 331 m | 329.5 m |
| Expected present-day elevation of past | 4 m | 331 m | 335 m | 333.5 m |
| MLC shorelines | 8 m | 335 m | 339 m | 337.5 m |
| **Model M4** | 0 m | 328.5 m | 335.5 m | 333.5 m |
| Expected present-day elevation of past | 4 m | 332.5 m | 339.5 m | 337.5 m |
| MLC shorelines | 8 m | 336.5 m | 343.5 m | 341.5 m |

*4.2. Chari River Delta Area*

In the area of the Dourbali lobe associated to the Chari river delta, the differential uplifts are 1.5 m, 2 m, and 3.5 m considering models M1, M3, and M4, respectively. In the M3 configuration (2 m of differential uplift), the coastal geomorphic features are expected to be found today at ca. 327 m for a water-level fitted to the Mayo Kebbi spillway crest, at 331 m if the thickness of the water overflow was of 4 m, and at 335 m if the latter was of 8 m (Table 3). Using model M1 does not significantly change these estimates, but following model M4, predicted elevation should be 328.5 m, 332.5 m, and 336.5 m, respectively (Table 3). Then, the strandplains lying in the 330–333 m elevation range on the Dourbali lobe of the Chari river delta may best reflect either a water-level for MLC at 4 m above the threshold elevation following model M3, or a MLC level adjusted at a slightly lower elevation in the model M4 configuration. A thick rather than thin lithosphere characterizing the Chari river delta area [46] suggests the former reconstruction (model M3, 4 m of overflow thickness at Mayo Kebbi) might be the best.

*4.3. North-Bodélé/Angamma Deltaic Complex*

Eight hundred kilometers to the north, herein referred to as the North-Bodélé/Angamma deltaic complex, a major delta system that drained the southern slope of the Tibesti Volcanic Massif developed [34], facing the deepest domain of the northern MLC sub-basin (Figure 1). This is the so-called "Angamma delta", often referred to on geographic maps as the "Angamma Cliff", the latter corresponding essentially to former delta foresets [29,33]. There, the strandplain lies in the 340–347 m elevation range and the late singular beach ridges correspond to a pristine longshore ridge, up to 5 m high, which has a maximum elevation of 341 m and thus reflecting a late shoreline at a slightly lower elevation, ca. 338 m. While the strandplain is truncated by a wide (>10 km) river channel network, the longshore ridge seals this channel network. As such, a basic cross-cutting relationship evidences that the MLC pertained in this area after the river stopped to flow [33]. In the North-Bodélé/Angamma deltaic complex area, the differential uplifts are of 4.5, 6, and 10.5 m following results of models M1, M3, and M4, respectively (Table 2). Whatever the considered overflow thicknesses (0, 4 or 8 m; Table 3), we argue that models M1 and M3 are not able to offer an acceptable elevation correction as none of the predicted shoreline elevations can fit the high elevations of the geomorphic features characterizing the northernmost shorelines of MLC. In the model M4 configuration (10.5 m of differential uplift), shoreline morpho-sedimentary features are expected to be found at 335.5 m for a water-level that was exactly adjusted to the Mayo Kebbi spillway, 339.5 m if thickness of the water overflow was 4 m, and 343.5 m if the latter was at 8 m (Table 3). Only the reconstruction considering 8 m of overflow

thickness would be here acceptable considering the main strandplain formation, yet this overflow thickness is poorly compatible neither with the Chari river delta area case study (see above), nor with the Goz Kerki spit area case study (see below). A slightly greater uplift than the one calculated with model M4 potentially occurred in the North-Bodélé/Angamma deltaic complex area, which would permit to form the beach ridges presently lying higher than 342 m with a conservative thickness overflow closer to 4 m than to 8 m. This assumption is reasonable considering the exceptionally thin lithosphere (<40 km) in the related area [46]. Last, in the Goz Kerki sand spit area where the strandplain lies in the 332–337 m elevation range, and the late singular beach ridge at around 330 m, similar considerations would conclude that model M4 offers a potential correction only in the case of low water-level adjusted to the spillway (Table 3), which is poorly compatible with the two other case studies. MLC reconstructions should then here rely on models with a relatively thick lithosphere (i.e., models M1 or M3), the best fits being obtain with an overflow thickness of 4 m, as far as the strandplain formation is considered. A slightly lower thickness of overflow would have occurred later for reconciling elevations of the singular beach ridges.

### 4.4. Kanem Dune Field Area

In the eastern Kanem dune field area (Figure 1), where the position of the MLC shoreline is poorly constrained, virtually no beach ridges or other constructive littoral features tied to former MLC shorelines are documented. Instead, a prominent erosion surface that truncates late Pleistocene aeolian barchanoid sand dunes developed during the subsequent MLC event [29,41]. A curvilinear erosional notch today positioned at 330 m delimits an eastern domain where the morphology of the massive dunes has been erased by waves (i.e., truncation of dune crests and infill of interdune depressions), from a western domain where the morphology of the dunes have been essentially preserved. In the eastern Kanem area, the differential isostatic uplift was ca. 4 m (in the 3–5 m range according to models M1–M4; Figure 3), suggesting that MLC's water-levels were coinciding in the Kanem area with the present-day 329, 333, or 337 m elevations when considering 0, 4, or 8 m of overflow thicknesses, respectively. Accounting for the hydro-isostatic uplift allows to position all the erosion surface in a subaqueous domain with a conservative overflow thickness at the spillway. If the lower estimate is, again, not in agreement with the 330 m elevation of the erosional notch, the two higher estimates suggest a few meters deep bathymetry existed along the notch, the latter being then itself a subaqueous feature rather than coinciding with the shoreline of the MLC. This suggests that the MLC waters penetrated farther westward, resulting in a labyrinthine wetland composed of partly flooded dunes, similar to the Bol Archipelago on the northeastern end of present-day Lake Chad. The large sedimentary work tied to the formation of this widespread erosion surface that developed at depths in the 3–15 m range suggests that it was coeval with the long-term MLC event related to the strandplain formation elsewhere around the lake [29,41]. No clear structure which would be coeval with the late, short-lived singular beach ridges is detected in this area.

### 4.5. Perspectives and Improvements

We here demonstrate that to integrate elevation correction based on the modelling of the hydro-isostatic readjustment allows the various lake-level elevations to be, at least at the first order, reconciled each together when considering a potential overflow through the Mayo Kebbi. This has been achieved in spite of (i) multiple combinations regarding the selection of Earth models parameters; (ii) unknown overflow thickness at the threshold—and the subordinate correction that should be applied if accounting for the additional load by a related 2–6 m thick water layer (Figure 4)—and (iii) various sources of uncertainties in estimating the mean water-level elevation from that of preserved coastal landforms (e.g., foredunes aggradation onto beach ridges, seasonal lake-level fluctuations, and variable wave climate).

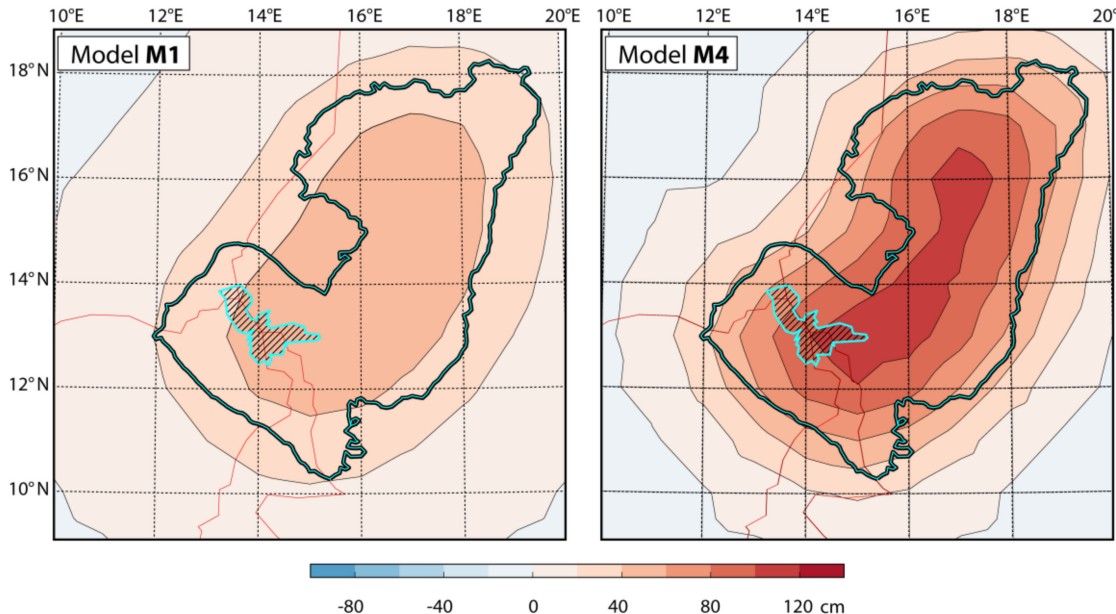

**Figure 4.** Vertical displacement (in cm) computed for an additional water load related to a 4 m thick overflow at the Mayo Kebbi spillway. Being mostly in the 0.5–1 m range, the subsequent correction that could be applied is arguably subordinate relative to the modeled total hydro-isostatic rebound (Figure 2) and thus has been neglected in reconstructions proposed in Table 3.

We put forward that a future MLC reconstruction that would minimize elevation differences from place to place will have to consider: first a mix of models M1–M3 and model M4 in area of thick (Dourbali-Chari, Goz Kerki) and thin (North Bodélé/Angamma) lithospheres, respectively, with weighted corrections in intermediate areas, and second a thickness of water overflow at the Mayo Kebbi threshold around 4 m, most likely one or two meters more at the time of the strandplain formation (12–6? ky BP), and one or two meters less in the later stage (around 5 ky BP) corresponding to the formation of the singular and unconformable beach ridges. This reconstruction would correspond to a paleo-lake which water-level was varying of ±2 m at time of MLC events (i.e., overflow thickness in the 2–6 m range), above a threshold at 323–324 m asl, and then imposing a water-level in the 325–330 m range. Related paleo-shorelines lie today at higher and different elevations: 1–2 m higher in the Dourbali-Chari area (326–332 m), 4–5 m higher in the Goz Kerki area (329–335 m), and more than 10 m higher in the north Bodele/Angamma area (>335 m). As such, there is no need to invoke fault-related neotectonic vertical movements for explaining differentials in shoreline elevation, that however could not be excluded in very particular places [66].

Further improvements would potentially rely on (i) the implementation of a more precise and complex initial water-loading model; (ii) a modelling permitting spatially-controlled lithosphere thickness changes, which are in the 40–100 km range over the study area according to Pérez-Gussinyé [46]; (iii) a more realistic and time constrained scenario for the onset, fluctuations, and termination of MLC, rather than considering the basic scenario of single and instantaneous drying up event. We however suspect that most of the expected elevation adjustments will remain within the error bars of the paleo-shoreline elevations as inferred only from the inspection of digital elevation models datasets, and will not change much the first order patterns described above. Clearly, hydro-isostasy impacted, and potentially still does, the lithosphere underlying the MLC. Its impact is significant and has to be accounted for when deciphering intricate schemes of shoreline elevation distribution at both the regional and local scales, as well as drainage patterns evolution over time.

## 5. Conclusions

The first modeling effort to quantifying the hydro-isostatic rebound related to Megalake Chad is presented in this study. A set of four different Earth models accounting for various thickness and viscosity of the lithosphere and the upper mantle, combined to a composite lake/load configuration have been tested with the software TABOO. Simulations have been simplified by starting at equilibrium, and by removing instantaneously the load due to the lake.

The uplift amplitude first and foremost results from the vertical displacement (several meters), whereas the change in the geoid height is very minor (some tens of centimeters). The uplift can reach 16 m in the deepest part of the paleo-lake and is in the 2–12 m range for its littoral domain. As such, the elevation of the paleo-shorelines is significantly impacted by the hydro-isostatic readjustment.

As a preliminary conclusion, integrating isostatic uplifts definitely improves the understanding of the spatial distribution of the paleo-shorelines of Megalake Chad, and possibly of those from other similar examples of megalakes (e.g., Aral Sea and Lake Eyre), though inconsistencies still pertain. This study indirectly reveals that the water-level of the lake when it reached its highstand was some meters above the elevation of its spillway.

Along with new simulations involving more detailed and complex models, the precise scenario of Megalake Chad's rise and demise has to be refined. This requires defining the most accurate lake-level indicators [42,67] and to systematically date their emplacement in order to later propose a comprehensive time and space reconstruction of this paleo-lake.

This work suggests that the vertical adjustment due to the unload of Megalake Chad is likely a still ongoing process, at least in the northernmost area. Since the termination of the African Humid Period was not an abrupt phenomenon [55] the regression of MLC must have lasted some time, as such the unload was not instantaneous as hypothesized in this study. Moreover, the subsequent aridification lead to an intensive aeolian deflation responsible for the massive removal of sediment away from the Chad Basin, representing another possible source for isostatic readjustment.

**Author Contributions:** Conceptualization, all authors; methodology, all authors; software and modelling, A.M.; validation, all authors; formal analysis, all authors; writing—original draft preparation, A.M., J.-F.G., J.H., and M.S.; writing—review and editing, A.M., J.-F.G., J.H., and M.S. All authors have read and agreed to the published version of the manuscript.

**Funding:** This research received no external funding.

**Acknowledgments:** Preliminary modeling was performed by A. Raingeard during an internship at EOST (University of Strasbourg). We are grateful to Ph. Paillou and T. Sternberg for inviting us to publish in this special issue, and for their great patience. We thank Jade Wei and Milica Djurdjevic for kind assistance alla along the publication process. This paper benefited of the insightful review made by two anonymous reviewers.

**Conflicts of Interest:** The authors declare no conflict of interest.

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
