# Peer review of "The Hydro-Isostatic Rebound Related to Megalake Chad (Holocene, Africa): First Numerical Modelling and Significance for Paleo-Shorelines Elevation"

_water, doi:10.3390/w12113180_

Round 1

Reviewer 1 Report

Reviewer:

Memin and colleagues provide an integrated analysis of hydro-isostasy in Megalake Chad (Africa) based on both numerical modelling and morphological-sedimentary features during the Holocene. Generally, the topic of the manuscript is of international interest, despite the limited spatial character of this work, the analysis and methodology are clear, and the manuscript is generally very well-structured. More explicitly, all data are sufficient, and the adopted methods are appropriated, as well as the treatment of the data. The figures are appropriated as both quantity and quality. The length of the paper is appropriated for this journal, with all interpretations and conclusions to be in general well justified. The text is also well organized, and this makes the manuscript easily readable and understandable. Finally the bibliography is accurate, without self-citations, and quite updated. Although I am not mother language, I could read the manuscript easily. The English is in relatively good shape, but some places need some improvements (please see my comments about rephrasing below). Overall, I have a couple of suggestions, and therefore ask for revision (minor) before accepting this manuscript for Water. So, please take them into account in order this promising contribution to be publishable. The manuscript is acceptable with minor revision.

Minor Comments and Suggestions:

-Abstract: The abstract is too lengthy. Try to reduce its length (for instance decrease the information of the Lake development and evolution in Lines 13-20) and mostly present your results.

-L41: References are needed here related with the climate change-Earth processes relationship

-L44-47:  Very long sentence and unclear! Please rephrase

-L52-56: Another example of a long sentence without clear meaning. Split it into 2 different sentences.

-L57: Replace “here” with “In the present study”

-L65: The authors could possibly use as comparison the drying of the entire Mediterranean Sea during the Messinian Salinity Crisis and the subsequent mantle resistance as a cause of this extreme geological event. For more information please see the work of Capella et al., 2019 (Terra Nova; 9 https://doi.org/10.1111/ter.12442).

-L75-77: Please rephrase this sentence

-L80: Replace “very large” with “huge”

-L81: The authors have to add references for Holocene Climatic Optimum and African Humid Period

-L82: “This climate event….”

-L 96: “…as key areas to assess the results….”

-L105-106: All these characteristics of the study area (e.g., Benue, Niger rivers) should be evident in Figure 1

-L111-112: Replace the word “some” with “about”

-L120: Where exactly is this evident? Mark them in the relevant figure

-L134: References are missing here

-L136-137: U and N should be in italics

-L149-151: It is unclear, please rephrase it

-L160: “…here divided into a northern and a southern sub-basin….

-L161: “The former being further subdivided…..”

-L175-184: This information is shown in Table 1 and should not be also described into the text. Delete the entire paragraph and keep the text concise without redundancies.

-L190-192: This is important. Please explain briefly the meaning of this in order to be understandable to the reader

-L204-205: Be specific by adding the uplift rates in numbers for both M2 and M3.

-L219-221: Describe this better.

-Discussion –General comment: The discussion is lengthy as it is and in some cases difficult to flow the text. For this reason I suggest to split it into sub-sections. For instance the authors could distinguish the different areas within the Lake setting as different subsections of the Discussion.

-L228-231: Not clear. Please rephrase

-L241: Delete the phrase “As a preliminary conclusion”

-L254: Explain how these heights (0, 4, and 8 m) were chosen

-L280-282: Please rephrase

-L323-326: References are missing for this sentence

-L328: Before this paragraph create a new subsection. The title could be “Perspectives and improvements” or something like that

-L328-336: Another example of a huge sentence which is very difficult to be understood by the reader. Split it into 2 or 3 different sentences.

Author Response

We are grateful for your encouraging general comments. We thank you for comments and suggestions which helped improving the clarity of this paper.

We considered with care all your comments and suggestions. We integrated almost all of these. Please see the details below.

Thanks for your time,

Mathieu Schuster

-Abstract: The abstract is too lengthy. Try to reduce its length (for instance decrease the information of the Lake development and evolution in Lines 13-20) and mostly present your results.

You are right, this was too long, therefore we reduced as suggested.

-L41: References are needed here related with the climate change-Earth processes relationship

Adequate references are given in the next few sentences.

-L44-47:  Very long sentence and unclear! Please rephrase

Done

-L52-56: Another example of a long sentence without clear meaning. Split it into 2 different sentences.

Done

-L57: Replace “here” with “In the present study”

Done

-L65: The authors could possibly use as comparison the drying of the entire Mediterranean Sea during the Messinian Salinity Crisis and the subsequent mantle resistance as a cause of this extreme geological event. For more information please see the work of Capella et al., 2019 (Terra Nova; 9 https://doi.org/10.1111/ter.12442).

This case study is really interesting, thanks for this reference. The paper by Govers et al. (2009) that is cited in Capella et al. is also very interesting for it relates hydro-isostasy to surface/landscape evolution, which has some links with our current paper. However, these papers are a bit out of the purpose of our paper which is focused on lakes and on the climate-induced isostatic rebound. The tectonic context from the paper by Capella et al. is really different and more complex than the one of the Chad basin which is an intracratonic sag basin. Therefore we prefer not cite this paper here, but we keep this in mind for our next paper.

-L75-77: Please rephrase this sentence

Done

-L80: Replace “very large” with “huge”

done

-L81: The authors have to add references for Holocene Climatic Optimum and African Humid Period

References are given in the cited paper, and moreover other classical references for this period are cited in the next sentences. The AHP being not the central topic of this paper, we decided to not emphasize on this topic, and references can be found in the papers that are cited here. As a consequence we added in the main text “and references herein”. The main idea here is that there was a major climate change that led to the rise and demise of a giant lake in the middle of the Sahara Desert, and the aim of this study is to quantify what was the response of the Earth’s crust to this load/unload stimulation.

-L82: “This climate event….”

done

-L 96: “…as key areas to assess the results….”

done

-L105-106: All these characteristics of the study area (e.g., Benue, Niger rivers) should be evident in Figure 1

I think that all names are reported in the map. Benue and Niger rivers were not displayed on this map because there was no room to do it, and moreover the drainage basin of the Benue and Niger rivers is immediately South/downstream of the “Mayo Kebbi spillway” which is well indicated on Figure 1. The sentence in the main text for this is “The basin of Lake Chad is endoreic up to the elevation of the topographic sill of the Mayo Kebbi (325 m asl) that connects to the south with the adjacent drainage basin of the Benue and Niger rivers (Figure 1).” We modified Figure 1 in order to include “Benue river” (lower-left corner), but not the Niger river which can not be seen on the map.

-L111-112: Replace the word “some” with “about”

Done

-L120: Where exactly is this evident? Mark them in the relevant figure

The map on Figure 1 allows to clearly see the Megalake Chad outlined by prominent paleo-shorelines. Zooming allows to see most of them, but all details can not be seen. Therefore, in order to be in line with your comment, we added in figure caption the following text: “For detailed views of the paleo-shorelines see Schuster et al. (2005 and 2014); Bouchette et al. (2010)”.

-L134: References are missing here

Right, reference 29 has been added

-L136-137: U and N should be in italics

Done

-L149-151: It is unclear, please rephrase it

done

-L160: “…here divided into a northern and a southern sub-basin….

done

-L161: “The former being further subdivided…..”

done

-L175-184: This information is shown in Table 1 and should not be also described into the text. Delete the entire paragraph and keep the text concise without redundancies.

I disagree, in the main text we explain why we selected these different models and their associated values. This is not changed.

-L190-192: This is important. Please explain briefly the meaning of this in order to be understandable to the reader

This sentence is supported by Figure 3, as well as by the next sentences.

-L204-205: Be specific by adding the uplift rates in numbers for both M2 and M3. Reference to Figure 3 has been added, from which values can be determined.

-L219-221: Describe this better.

Rephrased.

-Discussion –General comment: The discussion is lengthy as it is and in some cases difficult to flow the text. For this reason I suggest to split it into sub-sections. For instance the authors could distinguish the different areas within the Lake setting as different subsections of the Discussion.

Yes, excellent idea, thanks, this has been done, and gives the paper more clarity.

-L228-231: Not clear. Please rephrase

Slightly rephrased.

-L241: Delete the phrase “As a preliminary conclusion”

Yes, done

-L254: Explain how these heights (0, 4, and 8 m) were chosen

As specified in the text these values are "arbitrary", within a realistic range of values.

-L280-282: Please rephrase

Done

-L323-326: References are missing for this sentence

References 29 and 41 have been added

-L328: Before this paragraph create a new subsection. The title could be “Perspectives and improvements” or something like that

Great, I buy it! This makes the whole chapter much clearer.

-L328-336: Another example of a huge sentence which is very difficult to be understood by the reader. Split it into 2 or 3 different sentences.

Yes, I agree that one is a tough one! It has been split into two sentences, and reorganized accordingly.

Reviewer 2 Report

Good paper! I suggest to add the Geological Setting chapter by moving part of Materials and methods. This chapter would also provide description of structure (faults?) that contributed to the origin of Chad Megalake.

I have marked a few minor corrections in the PDF file

Author Response

Many thanks for your nice review and kind appreciation of the paper. Thanks also for the annotated pdf.

Your main comment is about reorganizing the "M & M" chapter. I kind of understand your point of view which makes sense. However, we decided to stick to the template provided by Water for the structure of the paper. As such, we keep as requested by the journal the description of the case study (ie Megalake Chad) in the chapter “Materials and Methods”, where a sub-chapter is dedicated. Furthermore, an expanded “geological setting” would be a bit out of the scope of this paper which we tried to keep short and focused. Information about the geological setting can however be easily found in several of the cited papers (eg Burke et al. 1977; Schuster et al. 2003, 2005 and 2009; and others that are cited in this paper).

Many thanks for your time and comments.

Mathieu Schuster